# How “Neuronal” Are Human Skin Mast Cells?

**DOI:** 10.3390/ijms231810871

**Published:** 2022-09-17

**Authors:** Magda Babina, Kristin Franke, Gürkan Bal

**Affiliations:** 1Fraunhofer Institute for Translational Medicine and Pharmacology ITMP, Immunology and Allergology IA, 12203 Berlin, Germany; 2Charité—Universitätsmedizin Berlin, Corporate Member of Freie Universität Berlin and Humboldt Universität zu Berlin, Institute of Allergology, Hindenburgdamm 30, 12203 Berlin, Germany

**Keywords:** mast cells, neurons, degranulation, dopamine, adhesion molecules, solute carriers, calbindin, transcription factors, skin, monoamine oxidase B

## Abstract

Mast cells are evolutionarily old cells and the principal effectors in allergic responses and inflammation. They are seeded from the yolk sac during embryogenesis or are derived from hematopoietic progenitors and are therefore related to other leukocyte subsets, even though they form a separate clade in the hematopoietic system. Herein, we systematically bundle information from several recent high-throughput endeavors, especially those comparing MCs with other cell types, and combine such information with knowledge on the genes’ functions to reveal groups of neuronal markers specifically expressed by MCs. We focus on recent advances made regarding human tissue MCs, but also refer to studies in mice. In broad terms, genes hyper-expressed in MCs, but largely inactive in other myelocytes, can be classified into subcategories such as traffic/lysosomes (MLPH and RAB27B), the dopamine system (MAOB, DRD2, SLC6A3, and SLC18A2), Ca^2+^-related entities (CALB2), adhesion molecules (L1CAM and NTM) and, as an overall principle, the transcription factors and modulators of transcriptional activity (LMO4, PBX1, MEIS2, and EHMT2). Their function in MCs is generally unknown but may tentatively be deduced by comparison with other systems. MCs share functions with the nervous system, as they express typical neurotransmitters (histamine and serotonin) and a degranulation machinery that shares features with the neuronal apparatus at the synapse. Therefore, selective overlaps are plausible, and they further highlight the uniqueness of MCs within the myeloid system, as well as when compared with basophils. Apart from investigating their functional implications in MCs, a key question is whether their expression in the lineage is due to the specific reactivation of genes normally silenced in leukocytes or whether the genes are not switched off during mastocytic development from early progenitors.

## 1. Mast Cells

Mast cells (MCs) are best known as potent effector cells of type-I allergic reactions, though they also act as early sentinels sensing pathogens and other perturbations of homeostasis [1,2,3,4]. The strategic location of MCs at host–environment interfaces close to blood vessels, nerves, and tissue-specific structures (such as hair follicles in the skin) enables the recognition of environmental factors such as allergens and venoms as well as endogenous mediators alike, including cytokines, neuropeptides, and alarmins [5,6]. The anatomic position and large spectrum of mediators that MCs can respond to and produce on their own enable them to integrate multiple signals and orchestrate succeeding responses [7,8]. Any aberrant regulation of MC activity can lead to hypersensitivity reactions or contribute to diseases such as allergic rhinitis, urticaria, angioedema, food allergies, asthma, and atopic dermatitis (AD) [9,10,11,12,13,14,15]. On the other hand, MCs are also deemed beneficial in several contexts such as the detoxification of venoms, host defenses against pathogens, and producing anti-cancer effects [16,17,18,19].

## 2. MCs and Evolution

Corroborating their beneficial nature in homeostatic (and presumably other) functions, MCs are evolutionarily old cells, appearing approximately 500 million years ago in invertebrates such as urochordates, which lack adaptive immunity [20,21]. Since their development precedes that of B cells and IgE, the lineage is believed to have important roles in non-IgE-driven processes. In fact, granular cells (the so-called test cells) were found in the chordate Styela plicata, surrounding oocytes and co-expressing heparin and histamine in their granules [22,23]. Similar cells were described in the Diplosoma listerianum, another ascidian [24]. These cells also contain tryptase–heparin complexes [20]. For a general overview of an evolutionary perspective on MCs, please see Ref. [21].

Interestingly, test cells are responsive to c48/80, the prototypical agonist of the MC-selective receptor Mas-related G-protein-coupled receptor X2 (MRGPRX2; Mrgprb2 in the mouse), and are thus believed to represent ancient effector cells of the innate immune system [23,24]. In fact, in the common salp, Thalia democratica, MC-like cells will release histamine, heparin, and TNF when exposed to bacteria [25]. Therefore, MC-like cells in urochordates are believed to organize the killing of pathogens, and act as inducers of inflammation, but also have a role in tissue repair [26]. Interestingly, test cells in the sea squirt Ciona intestinalis also produced prostaglandin D2, an entity expressed by few cell types (such as MCs and eosinophils), and they contained cDNA for an enzyme resembling human PGD2 synthase [27].

Ontogenetically, MCs were previously believed to derive solely from hematopoietic stem cells, but it is now established that MCs are of a dual origin [28,29]. In mice, a first wave of MCs originates from yolk-sac progenitors followed by a second wave of bone marrow-(BM-)derived MCs. Thereby, embryonic MC populations are gradually replaced by HSC-derived MCs. MC-committed progenitors circulate in blood in humans and mice and mature in peripheral tissues under the influence of several growth factors such as the stem cell factor [30,31,32]. MCs in certain tissues, especially the skin, can self-renew without replenishment by the BM, building so-called stable territories [33]. However, MC progenitors from the BM will enter the skin under inflammatory conditions and perturb the established territories [33].

## 3. MCs Communicate with Nerves

The MC–neuronal interaction has been briefly touched upon but was recently reviewed by others [7,34,35,36,37,38].

MCs are found in the central nervous system (CNS) and in peripheral tissues, including the skin. In the brain and spinal cord, MCs are mainly located in the dura mater and meninges, which play critical roles in the regulation of brain–immune interactions [36], but they can also infiltrate the brain, especially the thalamus [39]. Within the CNS, MCs are located on the abluminal side of the blood–brain barrier close to astrocytes and neurons [36]. CNS MCs may be involved in the control of pain [39,40], but they may also contribute to neuronal development and cognitive function [41,42].

It is documented that MC mediators can cause an increase in blood–brain barrier permeability [36]. In this regard, CNS MCs may play distinct roles in physiological and pathological conditions, and influence the course of neurodegenerative diseases, including Alzheimer’s and Parkinson’s disease; their involvement in the manifestation of migraines is especially well characterized [34,38,43].

The MC–neuronal connection in the periphery has been explored for decades and has been covered by many reviews. In fact, MCs are found in intimate contact with (mostly) sensory nerve fibers, and their numbers are particularly high in tissues innervated by small-caliber sensory A-delta and C-fibers, including the skin and visceral organs, where they act as important constituents of the neuroimmune interface [7,34,36,37,38,44].

On the one hand, MCs respond to neuronal products, on the other, mediators released by MCs, especially histamine and proteases, cause action potentials in neurons and modulate nerve fiber functions [34] (Figure 1). Moreover, sensory neurons express receptors for several MC-derived cytokines and employ similar signaling pathways as immune cells, so that actions from the nervous and immune systems can be orchestrated to generate concerted responses to danger signals [45] (Figure 1).

The MC–neuronal unit is especially relevant in the context of pain and, in the skin, more commonly in itch transmission through interactions with nociceptors and pruritoceptors, respectively [7,35,45,46,47]. Pruritus or itchiness is a leading symptom in many allergic and inflammatory skin pathologies such as atopic dermatitis, psoriasis, urticaria, prurigo, and others [35,46]. The roles MCs and their mediators play in itch transmission have been the subject of several excellent reviews [35,45,46,47]. It is of interest in this regard that MC granule mediators released by exocytosis comprise two typical pruritogens, namely, histamine and tryptase, which act by activating histamine receptors and PAR2 (and probably also Mrgprs) on sensory neurons. De novo synthesized cytokines such as IL-13 and IL-31 [48,49], which mainly act by the priming of sensory neurons, can also induce neuronal growth and sprouting [35,46,50] (Figure 1).

As a key element of the MC–neuronal unit, in the context of itching and beyond, the G-protein-coupled receptor MRGPRX2 has recently moved to the center of attention. Importantly, MRGPRX2 is the receptor for a long list of endogenous and exogenous ligands, including various neuropeptides such as cortistatin, somatostatin, SP (Substance P), VIP (vasoactive intestinal peptide), PACAP (pituitary adenylate cyclase-activating peptide), and others [51,52]. It will not be further covered herein, but the reader is referred to recent reviews on MRGPRX2 function in health and disease [5,53,54,55,56,57,58,59]. In contrast to MRGPRX2, which is mainly restricted to selected types of MCs, such as those in the skin, other members of the Mrgpr family are present on sensory neurons where they are involved in the transmission of itch responses and probably pain, including MrgprA1/C11 in the mouse, MRGPRX1 in humans, and MRGPRD in both species [45,57].

Of particular interest in the context of pruritic skin diseases is the crosstalk between sensory neurons, keratinocytes, and MCs, wherein TSLP (thymic stromal lymphopoietin), tryptase, epidermal proteases, and PAR-2 act as its crucial elements (Figure 1). PAR-2 activation, e.g., via tryptase-induced scratching behavior and the Ca^2+^-dependent release of TSLP, and TSLP directly activated a subset of TRPA1-positive sensory neurons to trigger itchiness [60]. Both subunits of the TSLP receptor, IL7Rα and TSLPR, are detected in a small subset of nociceptive neurons that do not overlap with either histamine or chloroquine-responsive neurons [45,60,61]. Skin irritation/scratching by itself gives rise to TSLP production in keratinocytes in a mainly IL-1α- and PAR-2-dependent manner—the latter likely through the activation of epidermal proteases [62,63]. Keratinocyte-derived and exogenous proteases also activate nerves (see above), which release neuropeptides such as SP. SP activates skin MCs for degranulation [64,65,66]. Interestingly, TSLP also primes MCs to respond more strongly to SP and other MRGPRX2 agonists (but not to allergens) [67], as it also maintains the survival of skin MCs [68]. Conversely, tryptase released from MCs further activates both keratinocytes (to produce more TSLP [69]) and sensory neurons [47] to initiate a vicious circle. In fact, it was observed that keratinocytes express some receptors that mediate pruritus when expressed on neurons, likely serving to establish feed-forward loops [46]. The same is also true for MCs and sensory nerves, the two being sensitive to the same mediators, including TSLP and NGF, implying a coordinated regulation of their functional outputs [67,68,70] (Figure 1).

Interestingly, MC granules can also be taken up by neurons in a process termed transgranulation, which occurs when MCs directly contact other cells to transfer granules into them [36]. In addition to MCs’ initiation or amplification of nociceptive and pruritoceptive stimuli, MCs have also been implicated in damaging axons, e.g., via extracellular traps [36].

Collectively, MCs and neurons are physically and functionally associated, and the MC–neuronal unit is involved in a number of processes, including pain, itchiness, and inflammation.

## 4. MCs Display Selective Neuronal Traits

MCs do not only communicate with nerves, but they also share properties with NS (nervous system) components, encompassing classical neurotransmitters and neuropeptides. In this regard, these selected MC subsets were reported to produce acetylcholine, gamma-aminobutyric acid (GABA), SP, VIP, PACAP, corticotropin-releasing factor (CRF), calcitonin gene-related peptide (CGRP), and neurotrophins such as nerve growth factor (NGF) and neurotensin, to name a few [44]. Moreover, MCs display receptors for neurotransmitters such as acetyl choline as well as neuropeptides [34], whereby a large number of the latter category is now known to signal via MRGPRX2 in MCs (in contrast to other cell types), as discussed in the previous chapter (Figure 1). Vice versa, MCs express substantial amounts of biogenic amines, especially histamine in humans and (in other species as well) serotonin, which both function as typical neurotransmitters; in fact, 50% of the histamine found in the brain has been reported to be stored in MCs [43]. In addition, MCs share similarities (albeit also differences) with neurons in their exocytotic machinery [71].

Further indications come from large-scale profiling efforts. Several omics studies have been carried out, including the body-wide Functional Annotation of the Mammalian Genome 5 (FANTOM5) atlas, a collection of nearly 1800 samples [72,73,74,75]. As reported in our FANTOM5 MC-focused paper, a striking cluster of genes expressed in MCs but absent from basophils were genes mainly expressed by neurons or the (central) nervous system (Supplementary Table S8; MCs > basos) [49]. This was further substantiated by a study establishing the most differential genes between MCs and other myelocytes, highlighting their exclusive expression in the lineage among leukocytes (Supplementary Table S1 [76]). This intriguing pattern was later confirmed by proteomic endeavors [77,78,79] (Ms in preparation). Genes associated with the nervous system were also detected in murine MCs [80,81]. So, while MCs are known to engage in an intimate crosstalk and form operational units with sensory neurons (Figure 1), it seems that MCs are also able to (re-)activate a portion of genes normally expressed in the nervous system (Figure 2). This relationship is further supported by the presence of neurotransmitters and neuropeptides in MCs, as briefly sketched above.

For this review, we extracted information from several “omics” studies and give a short overview of their pros and cons. In the FANTOM5 project, transcription start sites (TSS) were mapped genome-wide following quantification by cap analysis of gene expression sequencing (CAGE-seq). CAGE-seq is a technology that focuses specifically on the 5′ sequence of RNAs by sequencing the library of captured capped 5′-end cDNAs. Since single molecule sequencing was applied without the amplification by PCR, some lowly expressed transcripts might have been missed. The great advantage was that all the nearly 1800 samples were sequenced at the same laboratory on the same platform, permitting direct comparisons across samples. The genes selected for this review are graphically depicted in Figure 2 together with their most likely locations within the cell.

Another source of information for our review is the recently published single-cell RNA sequencing (scRNA-seq) data from healthy skin tissue (https://developmentcellatlas.ncl.ac.uk/datasets/hca_skin_portal/, accessed last on 1 July 2022) [82] (depicted in Figure 3).

scRNA-seq enables not only gene expression at the single cell level, but also the deciphering of cellular heterogeneity.. However, scRNA-seq methods still have some technical limitations. There are two major issues to consider when dealing with scRNAseq data. First, the method can only detect a limited number of mRNAs in a single cell. Second, processing the tissue into individual cells can result in high noise in the generated data [83]. Despite these limitations, the data provide highly valuable insights into tissue architecture and cellular heterogeneity. In addition, we used information from three mass spectrophotometric (MS) proteome analyses of skin MCs. They were performed by three different groups [78,79], including ours (Ms in preparation). MS-based techniques are used to study proteins on a large scale and can handle the complexities associated with proteomes. However, some limitations of MS-based proteomics should also be noted, as the identification of membrane-integral and associated proteins by mass spectrometry has lagged behind soluble proteins [84]. Since each technique has its limitations and boundaries, we incorporated and integrated data from transcriptomics and proteomics for a cross-comparison.

## 5. Purpose of Review

In this review, we discuss several genes that are overexpressed in MCs in comparison to the hematopoietic system by systematically bundling information from the above large-scale efforts, especially those contrasting MCs with other cells on the same platform. We focus on recent advances made with respect to human tissue MCs, but also refer to mouse studies whenever appropriate, considering that MC research has been traditionally performed mainly in rodents. Therefore, at the end of this overview, we additionally list the expression values of the genes we selected in mouse MCs. Many of these genes, which distinguish MCs from other leukocytes, are chiefly expressed in the nervous system. In this review, “neuronal” means pertinent to the nervous system, i.e., expressed in the nervous system in the first place (without the exclusion of other organs), including neural stem and progenitor cells, differentiating and mature neurons, neural crest-derived cells, oligodendrocytes, and astrocytes. In broad terms, “neuronal genes” expressed by MCs can be classified into those related to traffic/lysosomes/secretion, receptors/adhesion molecules, the dopamine system, Ca^2+^-related genes, and, as an overall principle, the modulators of transcriptional activity.

We discuss the members of some of the subcategories and give detailed descriptions of their roles in MCs and/or throughout the body if no information can yet be found for MCs. As a paradigm, we focus on the dopamine network, MLPH/RAB27, and the degranulation apparatus, as well as two other genes overexpressed in MCs against leukocytes, in all sources utilized (CALB2 and L1CAM). The reasons for the similarity between the NS and MCs will stem from regulatory networks further up the hierarchy, and the final section concentrates on the factors involved in the regulation of transcriptional activity.

## 6. The Dopamine System

Dopamine can be detected in the skin, for example, by dermal microdialysis [85], and its function has been documented in processes such as wound healing [86]. There is also ample support in the literature for a connection between MCs and the dopamine system. For instance, MCs have long been known to take up exogenous amines such as dopamine and serotonin, transport them to their specific granules, and release them again when triggered to degranulate, e.g., via c48/80 [87,88]. Dopamine storage also increased in the course of MC maturation from bone marrow precursors, and depended on the presence of serglycin [89]. However, in contrast to murine BMMCs (bone-marrow-derived MCs) [89], human skin MCs do not express tyrosine hydroxylase [49,72] and are therefore unlikely to synthesize dopamine, yet they may uptake the transmitter when produced by other cells. Further targets of dopamine located in the skin are melanocytes and fibroblasts [90,91].

In addition to dopamine internalization, MCs were reported already in the early 1980s to rapidly eliminate the transmitter independently of degranulation [92]. Murine BMMCs were also found to store relatively high levels of dopamine, while exogenous dopamine (20–100 µM) decreased proliferation and increased apoptosis, a process inhibitable by antioxidants, indicating that MC functions can be modulated by dopamine [93], reminiscent of what was later described for normal human dermal fibroblasts [91]. Inhibitory effects by dopamine and dopaminergic agonists were also reported with respect to the degranulation of RBL-2H3 cells (a rat MC line), whereby suppression occurred independently of early signaling events [94] (Figure 4). The inhibition was extended to cytokines, but the respective study employed a combined D1 antagonist/D2 agonist, which suppressed TNF release from rat MCs in a dose-dependent manner [95]. Following treatment with the same compound (GLC756), the lipid-derived mediator LTC4 was likewise attenuated in RBL-2H3 cells stimulated via FcεRI [96] (Figure 4). In another study, the D1-like receptor antagonist SCH 23390 suppressed mast cell degranulation (as well as Th2 cell differentiation) in a model of atopic dermatitis, suggesting that D1-like receptors facilitate immediate and late phase skin reactions by acting on T cells and MCs [97]. Combined, the data point to D1 as being a facilitator and D2 as a reducer of MC functions (Figure 4). A general overview of the differential roles of D1- and D2-dopamine receptors can be found in an excellent review [98]. The precise effects of dopamine on MCs will, therefore, partially depend on the ratio of different receptor subsets. In human skin MCs, the expression of DRD2 strongly exceeds that of DRD1, the latter being below detection, while DRD2 is highly expressed (Table 1) [49]. In accordance, dopamine can exert inhibitory effects in delayed skin hypersensitivity, such as the Arthus reaction (and centrally) [99]. Yet, other reports indicated that dopamine (taken by patients for hypotension) had no effect on wheal-and-flare reactions when induced by allergen or c48/80, but did so when triggered by histamine in the studied cohort [100]. The reason for the latter finding may hint at the differential effects of dopamine on MC degranulation versus dermal reactivity to histamine. This would imply that the degranulation of human skin MCs is not affected by dopamine, but to our knowledge, this has not been directly demonstrated so far. In addition, the effects are likely dose-dependent, and the effective pharmacological concentration achieved in the skin in Ref [100] may have been insufficient to influence cutaneous MCs.

In addition to the dopamine receptor DRD2, MCs express dopamine-degrading enzymes and transporters, as described below (Figure 4).

### 6.1. MAO-A and MAO-B

Monoamine oxidases (EC 1.4.3.4), namely, MAO-A and MAO-B, are isoenzymes that catalyze the oxidative deamination of neuroactive and vasoactive monoamines such as dopamine, adrenaline, and noradrenaline, as well as various xenobiotics [101,102,103]. Their genes are closely located in opposite directions on the X chromosome and share the same exon–intron organization, suggesting that they arose from the duplication of a common precursor gene [104]. Dopamine is a shared substrate, but MAO-A preferentially degrades serotonin and noradrenaline, while MAO-B favors phenylethylamine and benzylamine [105]. MAO-A is found primarily on the mitochondrial outer membrane (Figure 4) of adrenergic and dopaminergic neurons, whereas MAO-B is most abundant in serotonergic and histaminergic neurons and astroglial cells. In the brain, MAO-B expression in neurons is non-exclusive, while MAO-A can be found exclusively in some dopaminergic neurons [106]. The ratio of MAO-A and -B in the different brain regions seems more dominated by MAO-B, and there is also a shift with age whereby MAO-A is dominant in infants, while MAO-B steadily increases thereafter until senescence [107].

MAOs are also expressed in peripheral tissues including hepatocytes, leukocytes, platelets, and MCs [108]. In FANTOM5, MCs displayed the highest expression level of MAO-B, whereas MAO-A was barely displayed (Table 1). The expression of MAO-B in mast cells was even higher than in neuronal tissues [72,75]. Consistent with this pattern, MAO-B, but not MAO-A, was identified in proteomics studies of ex vivo mast cells (Table 2) [77,78]. Conversely, the expression of MAO-A was induced by the in vitro expansion of mast cells under treatment with SCF and IL-4 (Table 1 and Table 3).

The rapid clearance and breakdown of monoamines in the brain is pivotal for the smooth functioning of synaptic transmission, which plays a key role in modulating mood and emotion and the control of motor, perceptual, and cognitive abilities. In some psychiatric disorders such as schizophrenia [109,110] and depression [111], as well as in neurodegenerative diseases such as Alzheimer’s [112] and Parkinson’s disease [113], the balance between the anabolic and catabolic pathways of monoamines is disturbed. In such contexts, monoamine oxidase inhibitors can be a therapeutic option [114,115]. Interestingly, in a model of the oxidative perturbation of the heart, the degranulation of cardiac MCs could be inhibited by an in vivo treatment with MAO inhibitors, though this might have been an indirect effect [116]. The direct functional implication of MAO enzymes in MCs has not been studied so far but may well encompass dopamine degradation (Figure 4).

However, the high expression of MAO-B at the baseline and the inducibility of MAO-A in proliferating MCs may also be relevant in the context of histamine elimination. Released histamine can be inactivated either extracellularly by the oxidative deamination of the primary amino group by diamine oxidase (DAO) or intracellularly by the methylation of the imidazole ring by histamine N-tele-methyltransferase (HNMT) [117,118]. Although MCs do not express HNMT, an inactive methylation derivative of histamine, tele-methylhistamine, was detected in MCs, particularly in their granules [119]. This suggests that histamine released from MCs may be initially metabolized by HNMT from other cells, and the derivate subsequently transported to MCs for further breakdown by MAO-B.

In summary, the high expression of MAOs in MCs is intriguing and the enzymes could perform important functions in the cutaneous environment; therefore, their precise functions in MCs should be scrutinized in the future.

### 6.2. SLC18A2

SLC18A2 (Solute-Carrier Family 18 Member A2), also known as VMAT2, is a vesicular monoamine transporter that is expressed in neurons to translocate dopamine, norepinephrine, serotonin, and histamine into synaptic vesicles in an ATP-dependent manner [120]. Its proper function is essential to the correct activity of the monoaminergic systems that have been implicated in motor control, stable mood, and autonomic function. SNPs in this gene may be associated with some neuropsychiatric conditions, including schizophrenia, borderline syndrome, and others, whereas two SNPs in the SLC18A2 promoter region represent a protective factor against alcoholism [120]. VMAT2 inhibitors are a class of drugs that lead to the depletion of monoamines such as dopamine in nerve terminals. They are used to treat Huntington’s chorea or tardive dyskinesia [121,122]. Analogously to MAOs, skin MCs showed the highest expression of VMAT2 among the cell types and tissues analyzed in FANTOM5 (Table 1). A robust expression was confirmed by the proteomic data of ex vivo MCs reported by Plum et al. (Table 2). The reason for this pronounced expression is related to the accumulation of monoamines, especially histamine, in MC granules [123], as demonstrated in VMAT2-deficient mice [124]. Taken together, while also part of the dopamine system, the preferential expression of VMAT2 in MCs likely mirrors the vast production and storage of histamine by these cells. Possibly, the pharmacological inhibition of VMAT2 could have beneficial effects in MC-related diseases if side effects elsewhere in the body could be spared, e.g., by appropriate dosing.

### 6.3. SLC6A3

This gene, also named DAT1, encodes a dopamine transporter [125]. In fact, most of the known neurotransmitter transporters are encoded by the SLC6 gene family [125], whose members function as high-affinity, sodium-, and chloride-dependent solute transporters [126]. SLC6A3 is mainly expressed in the brain, especially in the substantia nigra [72,75,127]. The main function of this transporter is the reuptake of dopamine into presynaptic endings to stop further signaling. Loss-of-function mutations for this gene are associated with infantile Parkinson’s disease [128]. Its overall expression in FANTOM5 was low, identifying it as a low-abundance transcript. Notwithstanding, MCs were among the samples showing the highest expression next to substantia nigra. The role of this transporter in MCs has yet to be experimentally unraveled but may organize the uptake of exogenous dopamine by MCs (Figure 4).

In summary, MCs express several components related to dopamine metabolism, transport, and function at high levels. As mentioned, while skin MCs lack enzymes responsible for dopamine synthesis (tyrosine hydroxylase, tyrosinase, and dopa decarboxylase), they display the highest level of MAO-B at baseline and the inducibility of MAO-A; conversely, dopamine-producing skin melanocytes and fibroblasts lack these catabolic enzymes [49]. Moreover, skin MCs express the dopamine transporter, DAT (SLC6A3), and the mixed transporter, SLC18A2, and there is robust evidence for MCs’ ability to take up extracellular dopamine and store it in their granules [87,88,89,129]. Based on the specific combination of network components, we postulate that cutaneous MCs may function as dopamine scavengers by eliminating dopamine released by melanocytes, fibroblasts, or other cells through temporary storage in granules and/or catabolic degradation by MAOs (Figure 4). In this theory, MCs would reduce the dermal availability of dopamine to interfere with its paracrine and autocrine function, but they may also act in a timely shifted pattern of dopamine release (Figure 4). Further studies will be required to more precisely assess the position of MCs in the dopamine network.

## 7. MLPH, RAB27, and Their Contributions to the Secretory Apparatus

Degranulation is arguably the most specific MC function and secretory granules are the most selective hallmarks of the lineage [123]. While proximal signal transduction after FcεRI aggregation is fairly well understood [2,3,16,130], even though novel insights keep being uncovered [64], the events that occur at the level of granule transport, docking, priming, and fusion are less comprehensively characterized, especially regarding the precise involvement of distinct partakers that in principle (and in analogy with other cells) orchestrate or modulate MC exocytosis. The reader is referred to an excellent review on this topic [71]. Here, we focus on several entities of the degranulation machinery, which are abundantly expressed in MCs and exceed expression in other secretory cells. Evolutionarily, they play important roles in the regulation of the exocytotic apparatus of neurons.

Melanophilin (MLPH) represents a Rab effector protein that is involved in melanosome transport in melanocytes [131]. Consequently, mutations in MLPH give rise to a subset of the Griscelli syndrome characterized by hypopigmetation and by a silver-gray sheen of the hair [132]. Melanosomes are specialized lysosome-like structures and are thereby analogous to the secretory granules (“secretory lysosomes”) of MCs. MLPH serves as a link between melanosome-bound RAB27A and the motor protein MYO5A. Of note, the crystal structure of the effector domain of melanophilin complexed with Rab27B-GTP has been determined [133].

Rab GTPases are involved in all aspects of intracellular vesicle trafficking, where they are associated with different effectors when in their active, GTP-bound form [134,135]. Importantly, of the roughly 60 members, 24 are enriched in, or uniquely found in, the central nervous system (CNS) [136]. Rab27 occupies a specialized role and is conserved across metazoans, but is absent from yeasts or plants [137]. While vertebrates contain two isoforms, Rab27A and Rab27B, only a single isoform is present in worms and fruit flies, where it is abundantly expressed in neurons and regulates synaptic vesicle trafficking [137].

The interaction between Rab27A or Rab27B with MLPH and other constituents of the vesicle-transporting apparatus is complex and seems cell-type dependent. Its abundant expression in MCs clearly hints at a role of MLPH in granule traffic (Figure 5). Several studies have investigated these issues. In murine BMMCs, MLPH deficiency led to a hyper-secretory phenotype. The authors reported that while Rab27b facilitates degranulation, Rab27a has an opposing role and inhibits degranulation via a Rab27a/MLPH/MyoVa complex that stabilizes cortical filamentous actin [138]. In fact, the knockout of either component of the complex led to a more active secretion, while the destabilization of actin integrity phenocopied the hyper-secretory phenotype, highlighting that an intact filamentous actin network constitutes a barrier to granule discharge [138]. A dominant role of Rab27b over Rab27a was also reported by another study, whereby Rab27 entities countered the increase in granules exhibiting fast movement [139]. The authors concluded that Rab27b regulates the transition from microtubule to actin-based motility in MCs, thereby facilitating degranulation [139]. While Rab27a and b seem to have opposite functions in MCs, both use the same effector, i.e., Munc13-4, and Rab27a-Munc13-4 is the more important complex regulating exocytosis in cytotoxic T cells, NK cells, and platelets [140,141]. In fact, though originally assumed to act redundantly, later evidence revealed that Rab27A and Rab27B play different roles in selected types of secretion, including in MC exocytosis [137].

The Rab27 complexes regulate the transport of secretory granules by mediating the binding to kinesin [142], and together with Munc13-4, they are also responsible for the tethering/docking of secretory lysosomes at the plasma membrane (Figure 5) [137,143].

Skin MCs seem to express relatively small amounts of the inhibitory isoform RAB27A (maximum about 7 tpm = transcripts per million), while RAB27B expression was highest in MCs across all samples in the FANTOM5 atlas, reaching over 1000 tpm. Plausibly, RAB27B was also highly enriched in MCs versus myelocytes or versus basophils (Table 1). The enrichment over PBMCs was confirmed at the protein level (Table 4). Similarly, the expression of Munc13-4 (gene UNC13D) is prominent in MCs. Due to its key role in the exocytosis of multiple cells, it is not differential against basophils or myelocytes. In fact, neutrophils and eosinophils were among the highest expressors of UNC13D in the FANTOM5 atlas. However, the protein abundance in MCs was enriched over PBMCs in the study by Plum et al. (Table 2), and ample Munc13-4 levels are likewise detectable in our own study’s proteome (manuscript in preparation) (Table 3).

While RAB27A expression is poor in skin MCs, another negative regulator of degranulation, namely, RAB37, was found to be most highly transcribed in MCs vis-à-vis all other FANTOM5 samples (Table 1), though it was also highly expressed in many other leukocytes. It was reported for RBL-2H3 cells that Rab37 can bind to Munc13-4 and suppress degranulation by counteracting the vesicle-priming activity of the Rab27–Munc13-4 complex [144] (Figure 5). Of note, Rab37 is selectively phosphorylated on several sites upon SCF-mediated KIT activation in MCs [145].

## 8. Ca++ Signaling

### CALB2

Calbindin 2 (CALB2), also termed calretinin, for which the latter name is more commonly used, plays a major role in the signal transduction and orchestration of synaptic events in neuronal networks [146]. Mice deficient in calretinin display disturbances of motor coordination and other functions [147]. Its expression in MCs was highly differential against basophils and myelocytes, as it was also among the top 20 hits in MCs against PBMCs at the protein level (Table 2, Table 3 and Table 4). Accordingly, the prominent and selective expression in MCs was found by immunohistochemistry, e.g., in the buccal epithelium [148] as well as in schwannoma and neurofibroma tissue [149]. In addition, calretinin served as a sensitive and specific MC marker in the skin, which was able to distinguish MC-associated lesions from other potentially confounding skin lesions [150].

Together with calbindin-D28 (CALB1) and parvalbumin, calretinin forms part of a major family of EF-hand calcium-binding proteins of the nervous system [151,152]. Of its six EF-hand domains, four bind to calcium with a high affinity in a cooperative manner, while one is a low affinity site, and one is non-functional; accordingly, calretinin seems to display dual kinetic properties acting as both a slow and rapid calcium buffer [146].

In the neocortex, mostly nonpyramidal neurons (GABAergic interneurons) are positive for one of the three calcium-binding proteins and, in general, different types of cells stain positively for each of them in a non-overlapping manner [146,152]. Due to this expression pattern, individual members have been used to selectively target specific cell types [147]. Conversely, CALB1 and CALB2 are abundant in some sensory neurons, and their co-expression is more common in peripheral sensory neurons [151].

While MCs express high levels of CALB2, they lack CALB1; however, they also express some PVALB [49]. These proteins have obviously evolved to fulfill distinct tasks, acting as modulators of intracellular calcium transients either as calcium sensors or as calcium buffers depending on the context [146,153]. Therefore, it will be of great interest to delineate the functional spectrum of CALB2 in MCs, especially its role in degranulation.

## 9. Adhesion Molecules

### 9.1. L1CAM

The L1 cell adhesion molecule (L1CAM, also termed CD171) forms part of the L1 subfamily, which encompasses four members with similar structural organizations: the Close Homolog of L1 (CHL1), Neuronal Cell Adhesion Molecule (NrCAM), Neurofascin, and L1CAM itself [154]. L1CAM presents as a cell adhesion molecule with important functions in the development and morphogenetic patterning of the nervous system facilitating axon growth and synapse formation through attractant and repellent properties [155,156,157]. It binds to a spectrum of partners and engages in cell–matrix interactions as well as in homophilic and heterophilic cell–cell interactions, e.g., by binding to integrins such as alpha Vβ3 [158]. The L1CAM gene is highly conserved; it is already detectable in Caenorhabditis elegans and Drosophila [159].

In the human genome, L1CAM is located on the X chromosome, and its mutations can give rise to X-linked mental retardation syndromes, which encompass X-linked hydrocephalus, MASA syndrome, X-linked complicated spastic paraparesis, and X-linked corpus callosum agenesis, which have been collectively termed CRASH syndrome [160,161]. Several missense mutations cause milder phenotypes compared to those leading to the truncation of large portions or the absence of L1 [162]. L1 knockout mice phenocopy the human defects displaying developmental anomalies in the nervous system [163].

On the other hand, the aberrant expression of L1 has been linked to the development of human carcinomas [164], influencing DNA damage responses and bestowing cancer stem cell-like properties, including KLF4 (Krüppel-like factor-4) and CD44 (well-established markers of stemness) [154]. Interestingly, the association between L1CAM and “stemness” was also found in human embryonic stem cells, where its depletion led to the downregulation of stem cell-defining TFs, while its overexpression had the opposite effect [154].

In addition to mediating adhesive interactions, the binding of CAMs can affect the intracellular signal transduction machinery and provide cytoskeletal linkage [165,166]. It is therefore assumed that the distinct processes regulated by CAMs including L1CAM can be caused by their adhesive properties, their signaling properties, or both. For example, binding to the cytoskeleton may inversely control the neurite-stimulating activity of L1CAM [167]. Interestingly, its cytoplasmic/transmembrane portion can shuttle to the nucleus after the shedding of the extracellular domain, while in contrast, L1CAM expression can be induced by DNA damage [154].

L1CAM expression in FANTOM5 was highest in melanocytes, brain regions, and MCs. Even though its expression in lymphocytes has also been documented [168], most samples of memory and naïve T and B cells were below detection in the FANTOM5 atlas [72,75].

The proteome analysis by Gschwandtner et al. detected high levels of L1CAM in purified MCs, but also in the MCs of normal, psoriasis, and mastocytosis skin, revealing its characteristic of being a novel lineage identifier, wherein L1CAM appears even in the title of their report [77]. Despite the robust evidence of its prominent expression in MCs, L1CAM’s function in these cells is unexplored. We recently found that L1CAM is strongly phosphorylated on two serine residues upon SCF-mediated KIT-triggering [145]. In fact, L1CAM is a known substrate of ERK2, and phosphorylation seems to control the binding of L1CAM to ankyrin, which in the nervous system can promote synaptic stability [154].

Of additional interest, L1CAM was found to be upregulated in the cerebrospinal fluid of fibromyalgia and rheumatoid arthritis patients simultaneously with increased KIT, both considered pain-related proteins in this context; their concurrent elevation may signify an increased MC load in the pathologies [169]. Conversely, L1CAM was downregulated in malignant peripheral nerve sheath tumors compared with plexiform neurofibromas along with other markers of MCs (such as MC chymase, CMA1), hinting at a possible depletion of MCs in this type of malignancy, although L1CAM downregulation was suspected to result from a reduction of Schwann cells by the authors [170].

Homeobox (Hox) and paired box (Pax) proteins orchestrate L1CAM expression in the neural axis, while inactivity outside of the nervous system can result from suppression by the so-called neural restrictive silencer elements (NRSEs), whereby an intronic NRSE in L1CAM limits transcription to the nervous system [171]. Together with the relevant functional underpinnings, how such elements may be overcome in MCs will be a key question for future studies.

### 9.2. NTM

Neurotrimin (NTM), together with the limbic system-associated membrane protein (LAMP, coding by LSAMP) and the opioid-binding cell adhesion molecule (OBCAM), constitute an immunoglobulin superfamily (IgSF) subfamily of neural proteins that are glycosylphosphatidylinositol (GPI)-anchored adhesion molecules engaged in heterophilic and homophilic interactions; these proteins mediate multiple specific cell–cell interactions in the developing nervous system, facilitating or inhibiting neuronal projections depending on the cell type [172]. Heterophilic interactions can occur with the other family members [173]. NTM expression is regulated during the development of the CNS in mice and rats, with expression detected in the neurons of the thalamus, cerebellar granule cells, Purkinje cells in the hindbrain, in the olfactory bulb, neural retina, dorsal root ganglia, and the spinal cord, and in the adult, especially in cerebellar systems and at excitatory synapses, suggesting a function in their stabilization into adulthood [174,175]. It has an even wider expression in the human brain, while its expression is likewise higher during brain development than in the mature brain, being also stronger in brain tumors than in normal brain tissues [176]. Its expression in the spinal cord seems to be downregulated by spinal cord injuries, suggesting its regulation via afferent input [177]. Its expression (and estrogen-mediated regulation) in the myometrium seems to assist estrogen-induced sympathetic pruning, highlighting its role as a repulsive protein in this setting [178]. The NTM gene was associated with intelligence and IQ in a genome-wide association study [179]. In accordance, knockout mice show deficits in emotional and other types of learning [180,181]. In Schwann cells, NTM is part of a proliferation and migratory program countered by miR-182 in association with sciatic nerve injuries [182]. Apart from its role in synaptogenesis, NTM also promotes axonal fasciculation, i.e., the process of a growing axon adhering to another, forming axon bundles.

NTM was most highly expressed in the substantia nigra, pineal gland, and spinal cord in the FANTOM5 atlas, closely followed by MCs (Table 1). It is highly conserved both in its transcription initiation and coding regions [74].

Within the NS, LSAMP and NTM are often expressed in a complementary fashion, potentially interacting with each other [173]. Indeed, no expression of the other family members is found in MCs; thus, it is an attractive hypothesis that NTM on MCs, by engaging with NTM or LSAMP on DRG neurons, will enable or strengthen MC–neuronal interactions, perhaps explaining the close connection between the two cell types in many tissues, including the skin (see above). However, MC-NTM may also have cell-autonomous, adhesion-independent functions, as suggested by a recent study of cultured hippocampal cells [181]. The exploration of its roles in the lineage definitely deserves future attention.

## 10. Transcriptional Regulators

Lineage-specific regulatory transcription factors, which bind to promoters and enhancers, are key elements in the orchestration of cell-specific transcriptional programs, conferring lineage identity.

Of the several TFs, the entities found in the nervous system and MCs comprised LMO4, PBX1, and MEIS2; in addition, the epigenetic regulator EHMT2, a histone-lysine N-methyltransferase, likewise fulfilled the criteria. The entities will be discussed in detail below.

Although overexpressed at the transcriptional level in MCs (as confirmed by a skin scRNA-Seq (Figure 1)), the transcriptional regulators were below the detection level in the proteome reported by Plum et al. (Table 2), while PBX1, MEIS2, and EHMT2 were recovered in our own proteomic effort (Table 3) and the latter two also in the endeavor specified in Table 4. The less efficient capture of some proteins may be related to the fact that many TFs are moderately expressed and poorly soluble. This leads to the relatively low sensitivity of MS-based proteomics and can preclude the identification of TFs. However, slight differences in sample processing may have increased the solubility and therefore the recovery (Table 3 and Table 4).

### 10.1. LMO4

This factor belongs to the LIM-only (LMO) family of transcriptional regulators, which are cysteine-rich proteins comprising two zinc-binding LIM domains but lacking the DNA-binding homeodomain; therefore, they mainly act as docking sites for other factors, facilitating the assembly of multiprotein complexes [183]. LMO4 was identified in 1998 [184] and has been studied quite extensively in a variety of tissues and organs both with respect to homeostasis and pathological conditions. However, it has remained poorly investigated in MCs despite its description in high-throughput screens. Its gene is highly conserved at the transcript initiation region (TIR) and in exons [74].

In FANTOM5, LMO4 was highly expressed in the brain, testis, adipocytes, and MCs, and it was clearly differential in MCs against basophils and other myelocytes (Table 1, Supplementary Table S1) [72,75]. Its mRNA is also abundant in the MC line HMC-1 (own unpublished data), a highly immature, malignantly transformed MC subset [185,186], which implicates that LMO4 expression may begin early in MC development.

Functionally, LMO4 is involved in the regulation of Ca^2+^ fluxes as it activates ryanodine receptor 2 (RyR2) expression [187]. Other Ca^2+^ channels in the hypothalamus are also under the control of LMO4, thereby regulating neuronal excitability [188]. On the other hand, LMO4 can itself be regulated by Ca^2+^, indicating a circular relationship [189]. LMO4 is also involved in ATP signaling in neurons and is required for ATP to protect neurons via P2Y purinergic receptors (also found in MCs) from hypoxia-induced apoptosis [190]. In accordance, LMO4 enhances the granulocyte colony-stimulating-factor-induced signal transducer and the activator of transcription 3 (STAT3) signaling in neurons, and this effect is partially mediated by the sequestration of Histone deacetylases (HDACs) [191]. Interestingly, LMO4 has also been associated with dopamine D2 receptor signaling in the amygdala, where it contributes to cue-reward learning [192]. LMO4′s generally protective or homeostatic nature in the CNS is underlined by its reduced expression in Alzheimer’s brains [193] and its obesity-offsetting role through the maintenance of central leptin sensitivity and food intake control via the regulation of Ca^2+^ channels in the hypothalamus [188,194]. LMO4 expression is developmentally regulated in murine brains and is expressed early on during CNS development [184,193,195]. Mice with germline ablation of LMO4 die before birth and show defects in neural tube closure and skeletal patterning [183]. LMO4 is also involved in neural crest development, where it interacts with Slug/Snail (that is, the transcription factors Slug (SNAI2) and Snail (SNAI1)) [196,197].

Since Ca^2+^ is so crucial in MC granule discharge and transcriptional programs, this factor may play important roles in the MC lineage as a regulator of the Ca^2+^ machinery and beyond. This is further underlined by the following findings: LMO4 requires partners to bind to DNA such as LDB1 [198], which is highly expressed by MCs. In addition to LBD1 (and other known partners), LMO4 has been described to be able to build a complex with SCL/Tal1 and GATA2 in the spinal cord [199]; these TFs are not only abundantly expressed by MCs, but they are also regarded as master regulators of the lineage [49,200,201,202]. Therefore, a detailed exploration of LMO4 and its complexes in MCs may provide valuable insights into how MC transcriptional programs are selectively initiated and maintained against other myelocytes.

### 10.2. PBX1

PBX1 (Pre-B-Cell Leukemia Homeobox 1) encodes the evolutionarily conserved pre-B cell leukemia homeobox transcription factor, which contains a TALE (three amino acid loop extension) homeodomain [203]. It forms nuclear complexes with other TALE-containing proteins to regulate target gene expression by modulating the biological activities of the Homeobox transcription factor family (and other proteins) [204,205]. PBX1 may act as one of the relatively few so-called pioneer factors, i.e., factors able to bind to heterochromatin, facilitating DNA access for other proteins [203]. Its diverse functions are partially achieved through differential regulation by alternative splicing [206,207,208].

Among other processes, PBX1 orchestrates organ and limb axis patterning during embryogenesis; therefore, its deficiency in mice leads to embryonic lethality [209]. The drosophila ortholog extradenticle (exd) is likewise involved in the establishment of segmental identity [210]. During mammalian embryogenesis, its highest levels are found in neuronal tissues such as the brain, spinal cord, and ganglia [211]. Its role in mammalian neurogenesis is further supported by its expression in proliferating cells of the subventricular zone and their neuronal progeny in the olfactory bulb, while being undetectable in glial cells [212]. In fact, in adult olfactory bulb neurogenesis, PBX1 not only controls neurogenesis in progenitor cells but it also serves as a terminal selector for dopaminergic fate, repressing alternative cell fate decisions [208]. A connection with dopaminergic neurons is further underlined by its reduction in the substantia nigra of Parkinson’s disease [203,213]. PBX1 haploinsufficiency indicates that the correct PBX1 dosage is crucial for different processes including brain development in humans [214]. Its mutations can lead to severe defects such as congenital anomalies of the kidneys and urinary tract, external ear, branchial arch, heart, and genitalia, and they cause intellectual disability and developmental delay [215].

Together with Hox genes and MEIS1, PBX1 is also found in early hematopoietic progenitors and is downregulated during differentiation [216]. The knockdown of PBX1 gives HOXB4-overexpressing HSCs a competitive advantage, hinting at a negative role of PBX1 in early hematopoiesis [217]. Conversely, PBX1 is required for the self-renewal of HSCs and its absence results in the loss of stem cell maintenance factors and the premature expression of cell-cycle regulators instead [218]. A significant proportion of PBX1-dependent genes is associated with the TGF-(transforming growth factor)-β pathway that contributes to HSC quiescence [218]; TGF-β pathway genes are also very active in MCs [49].

The association of PBX1 with stemness is underlined by its activity at the Homeobox gene NANOG promoter, a key regulator of pluripotency, where PBX1 cooperates with KLF4 to maintain the undifferentiated state of human embryonic stem cells (hESCs) [219]. In fact, both PBX1 and MEIS2 (see next paragraph) can interact with KLF4 and be recruited to the respective GC box elements occupied by KLF [220]. The relationship between PBX1 and NANOG is complex, since NANOG also regulates the PBX1 promoter [221]. There is also a mutual relationship between PBX1 and AKT signaling, with PBX1 being both upstream and downstream of AKT activity [221]. In the latter study, PBX1 overexpression led to the increased phosphorylation of AKT accompanied by the proliferation of mesenchymal stem cells [221], which seems opposite from the findings in hematopoiesis, where quiescence was favored instead (see above). Rather, the expression pattern in MCs hints at a similarity with mesenchymal stem cells since higher levels of PBX1 are found in proliferating MCs than in their barely proliferating ex vivo counterparts (Table 1). While not specifically studied in MCs, a connection with KIT, the principal receptor tyrosine kinase of MCs, was recently reported, whereby KIT activation in ovarian cancer stem cells could stabilize Notch Receptor 3 (NOTCH3) and thereby increase the downstream target PBX1 [222]. MCs do not express Notch3; however, they do express other members of the Notch family. An investigation of PBX1’s role in MCs may identify important aspects of cell cycle regulation and the establishment of the molecular individuality of the lineage.

### 10.3. MEIS2

Another remarkable TALE-class homeodomain TF is Meis2 (Myeloid Ecotropic Viral Integration Site 1 Homolog 2), a highly conserved factor both in the exon structure and in its transcription initiation region, which are both in the first percentile of genes [74]. Its mRNA was highly differential in MCs against other blood cells (Table 1). In addition, Meis2 represented one of the most highly differential genes between mature PMCs and immature (in vitro-generated) BMMCs in mice [223] (see also forthcoming Table 5). The paralog Meis1 was detected in murine BMMCs where it bound to, among others, the KIT promoter, even though the Meis1 expression and genome-wide binding potential were substantially stronger in HSCs [202]. Meis1 is undetectable in human skin MCs [49], pointing towards Meis2 as the key player in these cells. As mentioned, Meis2 can form complexes with PBX1 and KLF4 to regulate gene expression. Of note, both PBX1 and MEIS2 were part of the MRGPRX2 cluster (genes overexpressed in MCs, especially in ex vivo MCs) in FANTOM5 (Figure 3 in [49]).

In line with its conservation, Meis2 plays key roles during development and cell fate specification, and it is spatially regulated in the developing nervous system; in an adult, its expression is mainly restricted to the brain and female genital tract [225]. More precisely, during early midbrain development, Meis2 expression is de-repressed by the removal of poly comb from a midbrain-specific enhancer [226]. Interestingly, Meis2 seems to facilitate the development of early hematopoietic progenitors from human embryonic stem cells, with stem cell leukemia/T-cell acute lymphoblastic leukemia 1 (SCL/Tal1) acting as downstream effectors [227]. SCL/Tal1 constitutes one of the lineage-defining TFs of MCs and is involved in the regulation of KIT expression [145,200,202]

Haploinsufficiency is associated with deregulated facial development, intellectual disability, and congenital heart defects [228,229,230], and the factor is also involved in adult olfactory bulb neurogenesis [231]. Meis2-deficient mice are embryonically lethal and show impaired cranial and cardiac development [232]. The defects result, at least in part, from perturbations of neural crest cells, as it was demonstrated using a crest-specific knockout [232].

It appears that the tissue-specific functions of Meis2 stem from interactions with the different binding partners. Various co-factors have been described, including Paired box protein 6 (Pax6) and Distal-Less Homeobox 2 (Dlx2), which combine with Meis2 during olfactory bulb neurogenesis [231], while Meis2 physically interacts with Pax3 and Pax7 during hindbrain development [233]. MCs express no Pax family members, yet they do express abundant levels of Pbx1 and KLF4, as mentioned in the previous paragraph, which are two factors co-operating with Meis2 at selected promoters [220]. In the hematopoietic system, Meis2 seems to be an early regulator of hematopoiesis at the level of the endothelial to hematopoietic transition through the regulation of several key transcription factors such as GATA-2, SCL/Tal1, and Growth Factor-Independent 1 Transcription Repressor (GFI1) [227]. These TFs are highly expressed in MCs and are implicated in MC development and function [49,130,202]. Interestingly, the enforced expression specifically of SCL/Tal1 rescues the defect in Meis2 during hematopoietic differentiation [227]. The preferential expression of Meis2 in MCs over other hematopoietic lineages may be related to their relative similarity with hematopoietic stem/progenitor cells [49,202], which seems somewhat paradoxical considering the high differentiation stage tissue MCs have attained, including their abundant expression of “MC-private” genes [49].

As with the other transcriptional regulators, insights into Meis2 function in MCs may uncover points of convergence in the TF networks of neurons and MCs, which may eventually explain why the otherwise unrelated cell subsets co-express a spectrum of genes absent from other hematopoietic lineages.

### 10.4. EHMT2

Epigenetic modifiers represent another crucial point in cellular differentiation, potentially bestowing lineage identity. Euchromatic histone-lysine N-methyltransferase 2 (EHMT2, also named G9a) is involved in histone post-translational modification. It represents one of the two evolutionarily conserved EHMT enzymes that specifically mono- and dimethylate the Lys-9 of histone H3 (H3K9me1 and H3K9me2, respectively).

EHMT2 was found to be overexpressed in MCs versus both basophils and myelocytes (Table 1), and it was also found in our proteome (Table 3), though it was below detection in Plum et al.’s study (Table 2). Its specific functions in MCs have not yet been delineated. Conversely, its functional implication in the immune system has been documented for lymphoid cells, in which memory establishment also requires specific gene repression (in addition to gene activation) to maintain a quiescent state [234]. Its expression in the HMC-1 cell line was comparable to that observed in skin MCs (own unpublished data), suggesting a stable expression in the lineage from immature to mature stages. Interestingly, however, EHMT2 is strongly downregulated in MCs following FcεRI crosslinking [49] and may contribute to the epigenetic changes that occur in activated MCs [235].

The specific tag H3K9me created by EHMT2 leads to epigenetic transcriptional repression through the recruitment of heterochromatin protein 1 (HP1, encoded by Chromobox 5 (CBX5)) to appropriately methylated histones [236]. CBX5 expression is present in MCs, but at an intermediate level (it is differential against basophils [49] but not against a pool of different myelocytes) [76]. The other methyltransferase, EHMT1 or GLP [236], is also expressed by MCs at relatively high levels, but the latter is highly expressed across leukocytes in general, including monocytes and basophils. The two proteins form heterodimers [234,237]. However, targeted deletions of either G9a or GLP revealed differences in their function, likely resulting from the unequal requirement of each subunit in the modulation of gene expression [234]. Additional context-dependent binding partners may dictate substrate specificity, especially that of non-histone proteins, which can also be modified by the complex [238]. For example, the EHMT-dependent methylation of p53 reduces its transcriptional activity [239]. On the other hand, the loss of G9a-dependent H3K9me2 stretches seems to confer a growth or survival advantage in cancer [239]. Accordingly, H3K9-selective methyltransferases, including EHMT2, have multi-faceted and divergent roles in cancer development [240].

A further layer of complexity is present in the fact that depending on the context, EHMT2 is also able to activate gene expression in a methyltransferase-independent fashion, probably by acting as a scaffold to recruit transcriptional coactivators such as Histone Acetyltransferase P300 (EP300) [234]. Regarding MCs, the described interaction of EHMT2 with the transcriptional repressor GFI1 is noteworthy, since MCs express the highest levels of GFI1 among non-transformed cells [72,75,241], and its expression is also differential against basophils and myelocytes [49,76]. GFI1 is involved in the silencing of multiple genes through the recruitment of repressive modulators including methyltransferases [234]. Indeed, GFI1-deficient and G9a-deficient mice show various overlaps in the immune system [234].

EHMT2 has a role in Th cell specification as a key component driving Th2-cytokines such as IL-4, IL-5, and IL-13 [234]. The support of Th2 cell functions is independent of its methyltransferase activity, and is duplicated in the innate lymphoid cells (ILCs) lineage, whereby IL-5 and IL-13 transcription in ILC2 is likewise controlled by the enzyme [234]. Accordingly, G9a restricts the development of Th17 and Treg cells. Since MCs in humans are also significant producers of IL-5 and IL-13 (but not IL-4) [48,242,243,244], it will be crucial to elucidate whether Th2-like responses of MCs likewise require EHMT2. Furthermore, it will be of relevance to study the genome-wide distribution of H3K9me1/2 marks in the lineage under resting and activated conditions, as these marks may well contribute to MC identity. An exploration of a possible interaction and communication between EHMT2 and GFI1 will likewise be of great importance.

## 11. Expression of the “Neuronal” Genes in Focus in Murine MCs

We employed data from the Immgen consortium to assess the gene expression levels in murine MCs. Such a comparison was important because the mouse is still the most widely employed species in MC research. The Immgen atlas contains a large variety of immune cells, but since it focuses on immune compartments, no parallel information on the nervous system is available. Notwithstanding, the datasets allowed for a direct comparison of MCs with basophils and eosinophils (Table 5).

The well-defined MC markers Cma1 and Mrgprb2, as homologs of human CMA1 and MRGPRX2, were strongly expressed in murine MCs from all five tissues but only weakly in basophils and eosinophils, as expected (Table 5). The other genes showed a mixed pattern that was overall less skewed towards MCs compared to the human system. A substantial overexpression of MCs was only found for five genes: Maob, Slc18a2 (its high expression in basophils was expected because it is also the histamine transporter), MLPH, Rab27b, and Meis2, suggesting that granule transport and biogenic amine scavenging may be conserved between mice and humans. Conversely, several genes were either more highly expressed in murine basophils than in MCs (L1cam and Lmo4) or were comparable between the two subsets. Eosinophils also showed a greater expression of several genes in the mice. This indicates important species-specific differences between mice and humans, whereby MCs and basophils may bear a greater resemblance in the mice. This could be related to an observation made in fish, where the morphology of MCs was between mammalian MCs and basophils [21], suggesting that MCs and basophils originally developed from a common precursor, only later adopting specific functions. The distinction would therefore be less sharp in mice (where bifunctional MC/basophil progenitors were spotted [30]) and more finalized in humans, where MCs and basophils form well-separated clades [49]. Accordingly, human skin MCs were deprived of several receptors important in innate immunity, including TLRs, which were maintained by basophils [49,78], perhaps because other innate immune cells took over the functions that required TLR expression. Conversely, skin MCs upregulated other receptors (including MRGPRX2), by which they were facilitated in order to communicate more intensely with sensory neurons (Figure 1).

## 12. Interconnections between Our Selection of “Neuronal” Genes

To predict the interactions of the neuronal genes discussed above with other genes transcribed in MCs, we performed a network analysis using the STRING database (https://string-db.org, accessed on 1 July 2022) [245]. This resource allows for a stepwise addition of further nodes to visualize their connection with the list of genes initially defined. Herein, up to the fourth step, the nodes were added to the basal model until interconnectivity was achieved. The final network is clustered to a specified number of clusters according to its topological properties (Figure 6). The finally created network contained three subnetworks (Figure 6). Hereby, dopamine-related proteins formed the first subnetwork (green), while a second subnetwork was created by transcriptional regulators (red). Proteins related to the degranulation machinery were connected separately and built a third segregated subnetwork (blue).

We repeated this type of analysis on the GeneMANIA web server with similar results (Figure 7) [246]. Among the proteins predicted by the two algorithms to be linked to our predetermined genes, we highlight a few interesting examples, namely, the β2 adrenergic receptor (ADRB2), Nuclear receptor subfamily 4 group A member 2 (NR4A2), β-arrestin-2 (ARRB2), and Neuronal calcium sensor 1 (NCS1), as they were likewise overexpressed in MCs.

ADRB2 belongs to the G protein-coupled receptor (GPCR) family and primarily binds to epinephrine. Interestingly, ADRB2 inhibited the IgE-mediated release of histamine from lung mast cells in a concentration-dependent manner and this contributes to its beneficial effect in the therapy of asthma patients [247]. While agonist or antagonist effects have—to our knowledge—not been studied in MCs of a skin origin, an agonist was shown to improve wound healing and reduce scarring in the skin [248]. According to the FANTOM5 dataset, ADRB2 is expressed in the skin predominantly by MCs, followed by keratinocytes, whereas it is absent from dermal fibroblasts. The precise functions of ADRB2 specifically in skin MCs will require further investigation.

NR4A2 (also known as Nurr1, RNR-1, NOT, and HZF-3) is a transcriptional regulator and member of the nuclear receptor superfamily. Ex vivo MCs rank first among NR4A2-expressing tissues and cell types, whereas their cultured counterparts display downregulation [76]. The activation of cultured human skin MCs by IgE receptor cross-linking reversed the downregulation and resulted in the high expression of NR4A2 and NR4A3 [76]. The same expression characteristics were observed in mouse BMMCs and human LAD2 cells [249].

Interestingly, NR4A2 is essential for neuronal differentiation, survival, and the maintenance of dopaminergic neurons, and the TF directly controls the expression of dopamine-related genes such as SLC6A3, SLC18A2, DRD2, and thyroxine hydroxylase, the enzyme responsible for dopamine synthesis [250,251]. Therefore, its pronounced expression in MCs makes immediate sense, and it will be a key question for future studies whether this TF contributes to the transcription of neuronal genes in MCs. Interestingly, NR4A2 also exerts anti-inflammatory effects by the trans-repression and clearance of the p65 subunit of NF-kappa-B (RELA), which targets inflammatory gene promoters in a signal-dependent manner [252]. Since many immune processes can be controlled by NR4A2 [253], research into this TF in MCs will likely unearth links to neuronal networks on the one hand and inflammation/anti-inflammation on the other hand.

β-arrestins were initially discovered in relation to GPCR desensitization, followed by internalization, but they can induce or facilitate signal transduction on their own [254]. In the former scenario, internalized arrestin–receptor complexes can migrate to intracellular endosomes where they may be uncoupled from G-proteins. Similar to ARRB1, ARBB2 is expressed in large amounts in nervous tissue and may regulate the function of synaptic receptors [255,256]. Both ARRB1 and ARRB2 are also expressed in MCs. We recently showed that they restrain MRGPRX2-triggered degranulation and ERK1/2 activation in human skin mast cells and organize ligand-dependent MRGPRX2 internalization [257]. Further research is needed to delineate how β-arrestins regulate GPCR signaling in MCs at the level of MRGPRX2 and beyond, including ADRB2 and DRD2.

Another interesting GPCR regulator detected by the network analysis was NCS1 (known also as Frequenin), which is expressed in both cultured and ex vivo MCs and further upregulated by IgE receptor cross-linking. It functions as a calcium-ion sensor to modulate synaptic activity and neurosecretion [258]. Accordingly, it may also regulate IgE receptor-triggered exocytosis in cultured mast cells [259]. It will be of interest to determine whether NCS1 has a role in the MRGPRX2-pathway of MC degranulation. Collectively, network analyses can highlight interesting candidates linked to our predefined set of neuronal genes, which without reaching the predetermined overexpression (in MCs against leukocytes) could operate in tandem through physical and/or functional interactions with the entities discussed in this review.

## 13. Summary, Future Perspectives, and Concluding Remarks

Focusing on the markers overexpressed in (skin) MCs against the backdrop of other leukocytes, we identify MCs as high expressors of several neuronal genes. Even when present in other immune cells, expression is mostly less pronounced, thereby discriminating MCs from other myelocytes. A rational combination of recent high-throughput datasets with published evidence of their functions in other lineages and organs can guide us towards a better understanding of the molecular underpinnings of this fascinating cell type, providing fresh insights into MC ontogeny and phylogeny. From the perspective of our selected genes, the preferential expression in MCs over basophils/eosinophils seems less pronounced in mice, but hints at differences between mice and man. This aspect should be elaborated further to understand in what contexts mice can serve as model systems of human MC pathophysiology.

Of the entities covered herein, no experimental research has yet been conducted on MCs with respect to several broadly studied ones, while for others, there is limited documentation present and mostly from immature MC models. This further emphasizes that our understanding of the basic principles underlying lineage specification, mastopoiesis, and MCs’ function is still rather rudimentary despite the considerable progress in recent decades. Therefore, the dentification of their specific roles in MCs constitutes a crucial focus for future work. It may be speculated, though, that neuronal gene expression in MCs may be related to the ancient nature of the cells, and there is evidence they are more strongly expressed in “constitutive MCs” (i.e., β7^Low^ MCs) vis-à-vis “inducible” MCs (β7^High^ MCs), the two subcategories detected in mice [80,260]. In fact, most of the genes selected by us are found in constitutive mouse MCs (Table 5), even though they discriminate murine MCs less strongly from other leukocytes. That a similar distinction based on expression β7 exists in humans is supported by the FANTOM5 atlas, since ITGB7 (the gene encoding the β7 integrin chain) is basically undetectable in skin MCs ex vivo. However, it is increased upon in vitro expansion during which MCs become more inflammatory (expressing greater levels of inflammatory cytokines for instance), while they downregulate MRGPRX2 expression [48,66,261,262]. In mice, β7^Low^ MCs are likewise the less inflammatory cells, but more reactive via Mrgprs [80,260]. In accordance, ancient “test cells” and MCs in fish are responsive to substance P and/or compound 48/80, which are ligands of Mrgprs or alike, thereby bearing characteristics of constitutive or connective tissue type MCs (in contrast to inducible or mucosal MCs). During a pathogen challenge, constitutive MCs may be efficient at luring neutrophils into tissues as with their ancient precursors in other species such as fish [21]. If we assume that neuronal genes are more strongly expressed in more ancient MCs, we propose a beneficial role at least for some of the genes, with dopamine responsiveness, timely shifted dopamine release, and dopamine scavenging roles being among them (Figure 4). In addition, the genes of the granule transport machinery (Figure 5) are required for MC degranulation, a function already encountered in “test cells” and other precursor cells and required in the battle against intruding pathogens. Interestingly, inducible mouse MCs (β7^High^) are rather hypogranular cells [260].

Future analyses of the roles of the selected genes will not only advance basic research but also help illuminate the specific contributions of MCs to pathologies versus health-maintaining processes and develop selective biomarkers for such conditions. Since MCs have been maintained for long periods [21], positive evolutionary pressure has evidently been exerted on the lineage. MC–neuronal communication is a hot topic at the moment, (as elaborated in the respective chapter above) and it seems plausible that MCs communicating intensely with sensory neurons also display neuronal genes at higher levels than cells with fewer connections since evolutionarily they seem to have developed as a unit. We also postulate that “test cells” and other phylogenetic precursors of MCs express(ed) “neuronal” genes, since the genes typically show a high degree of conservation and could have been expressed in the developing nervous system and ancient “MCs” in tandem. However, we need to emphasize that both of the above postulates are speculative for now.

Transcriptomics and proteomics were able to highlight MCs’ particularities compared to (other) leukocytes, and further comparisons with evidently less related systems (such as neurons) can be viewed as a powerful tool to eventually help reveal the drivers behind lineage identity specification, maintenance, and subtype generation, as intended herein. While ontogenically distinct from neurons, MCs seem to be endowed with the required machinery to (re-)activate (or never repress) a selective portion of neuronal genes. An important issue will involve the way in which these genes are activated in the nervous system and in MCs to understand how the latter are able to “hijack” or repurpose them. Since most neuronal genes are inactive in MCs, as would be expected, the question is what distinguishes the active from the many silent ones. Of note, and as indicated in the above paragraphs, most of the genes described herein are highly conserved in evolution [75], as can be examined through the FANTOM-CAT webpage (https://fantom.gsc.riken.jp/5/suppl/Hon_et_al_2016/vis/#/, accessed on 1 July 2022). It will also be important to explore the expression of their orthologs in test (and other MC-like) cells in primitive organisms, as this strategy may provide further insights into the evolutionary origin of MCs. Collectively, in addition to the functional implications of gene (re-)expression (or lack of repression) in tissue (especially skin) MCs, insights into the machinery behind this process will be intriguing to understand molecular MC programming.

## Figures and Tables

**Figure 1 ijms-23-10871-f001:**
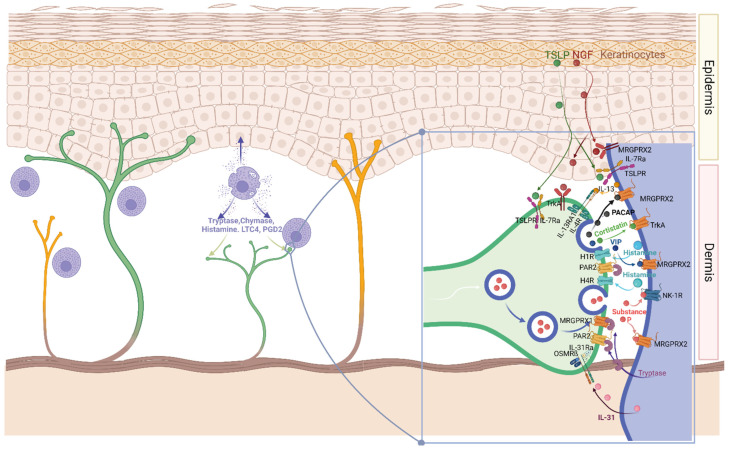
MCs communicate with sensory nerves in the skin. MCs are frequently situated near neurons, whereby the two are engaged in a close dialogue. Neuropeptides such as SP and cortistatin degranulate MCs, chiefly by activating on MRGPRX2; NK-1R can be contributory in the case of SP. Mediators exocytosed from MCs comprise the pruritogens histamine and tryptase, which act on H1R, H4R, and PAR2 on nerve endings (and potentially Mrgprs, i.e., MRGPRX1) on sensory neurons. In addition, the MC-derived cytokines IL-13 and IL-31 can further prime sensory neurons and induce their growth and sprouting. In the skin, keratinocytes occupy an important role and strongly influence both neurons and MCs, e.g., by provision of NGF and TSLP. See text for further details. For the sake of simplicity, the communication is depicted for one nerve only; however, in reality, different receptor combinations characterize subsets of sensory neurons, whereby distinct entities exclude others. Abbreviations: H1R (Histamine H1 receptor), H4R (Histamine H4 receptor), IL-13 (Interleukin-13), IL-13RA1 (Interleukin-13 receptor alpha 1), IL-31R (Interleukin-31 receptor), IL-7Ra (Interleukin 7 receptor alpha), LTC4 (Leukotriene C4), MRGPRX1 (Mas-related G-protein-coupled receptor member X1), MRGPRX2 (Mas-related G-protein-coupled receptor member X2), NGF (Beta-nerve growth factor), NK-1R (Neurokinin receptor-1), OSMRß (Oncostatin-M-specific receptor subunit beta), PACAP (Pituitary adenylyl cyclase-activating protein), PAR2 (Proteinase-activated receptor 2), PGD2 (Prostaglandin D2), TrkA (Tyrosine kinase receptor A), TSLP (Thymic stromal lymphopoietin), TSLPR (Thymic stromal lymphopoietin receptor), and VIP (Vasoactive intestinal peptide).

**Figure 2 ijms-23-10871-f002:**
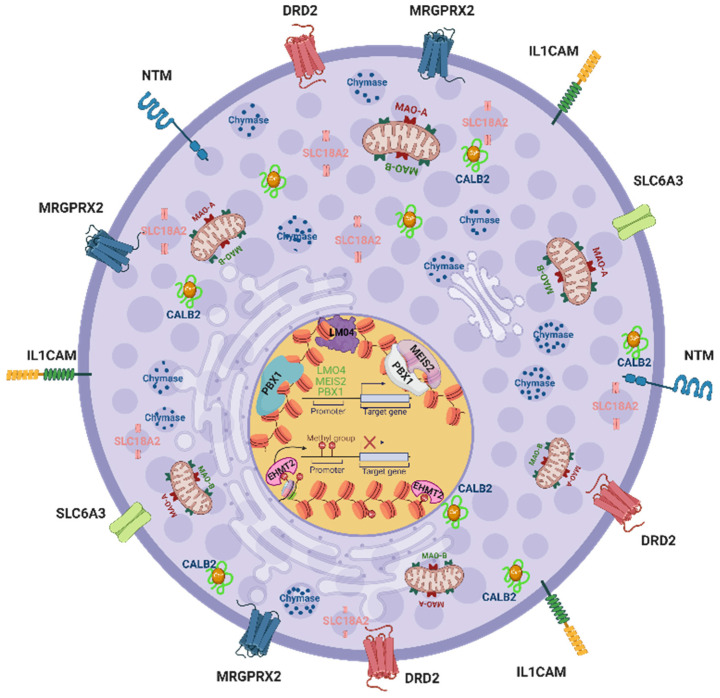
The nervous system-related genes covered in this review. A unifying element is their preferential expression in MCs in comparison to other myeloid cells. Depicted is the principal cellular localization of the following entities: DRD2, MAOA, MAOB, SLC18A2, SLC6A3, MLPH, RAB27B, RAB37, CALB2, L1CAM, NTM, LMO4, PBX1, MEIS2, and EHMT2. The genes MC chymase (CMA1) and MRGPRX2, expressed only by MCs and not by other constituents of the body (according to the comprehensive FANTOM5 atlas), have been added for comparison. Abbreviations: CALB2 (Calbindin 2 or Calretinin), DRD2 (Dopamine Receptor D2), EHMT2 (Histone-lysine N-methyltransferase), L1CAM (Neural cell adhesion molecule L1), LMO4 (LIM domain transcription factor 4), MAOA (Amine oxidase (flavin-containing) A), MAOB (Amine oxidase (flavin-containing) B), MEIS2 (Myeloid-Ecotropic Viral Integration Site 1 Homolog 2), MRGPRX2 (Mas-related G-protein-coupled receptor member X2), NTM (Neurotrimin), PBX1 (Pre-B-cell leukemia transcription factor 1), SLC18A2 (Synaptic vesicular amine transporter), and SLC6A3 (Sodium-dependent dopamine transporter).

**Figure 3 ijms-23-10871-f003:**
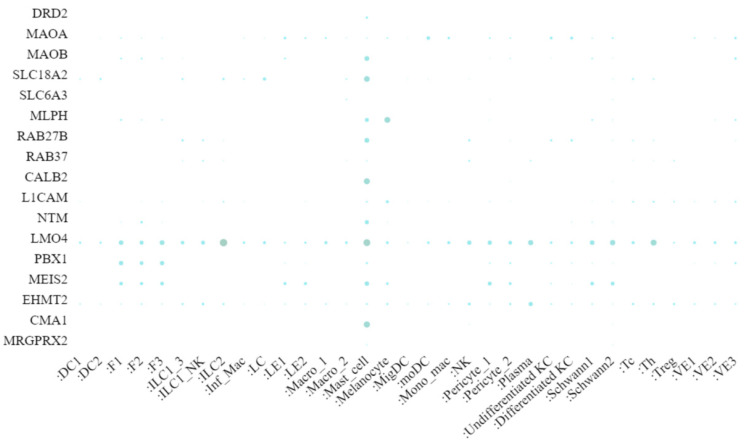
scRNAseq data of healthy adult skin. The expression of selected genes was extracted from scRNAseq analyses of healthy adult skin and is presented as dot plots; the data are from Reynolds et al. [82]. The color-coded size of the points corresponds with the expression level (the larger and darker, the more highly expressed). (https://cells.ucsc.edu/?ds=healthy-human-skin, accessed last on 1 July 2022) In the associated study [82], the transcriptomes of 200,462 healthy, single skin cells, which were classified into 34 different cell types, were sequenced using the scRNA-seq platform (10× Genomics); the link to the atlas can be found here: https://cells.ucsc.edu/?ds=healthy-human-skin, accessed last on 1 July 2022. Abbreviations: DC, dendritic cell; F, fibroblast; Inf_Mac, inflammatory macrophage; ILC, innate lymphoid cell; KC, keratinocyte; LC, Langerhans cell; LE, lymphatic endothelium; Mono mac, monocyte-derived macrophage; Mig., migratory; MoDC, monocyte-derived dendritic cell; NK, natural killer cell; Tc, cytotoxic T cell; TH, T helper cell; Treg, regulatory T cell; VE, vascular endothelium. The numbers 1, 2, and 3 represent different states or subtypes of cell types.

**Figure 4 ijms-23-10871-f004:**
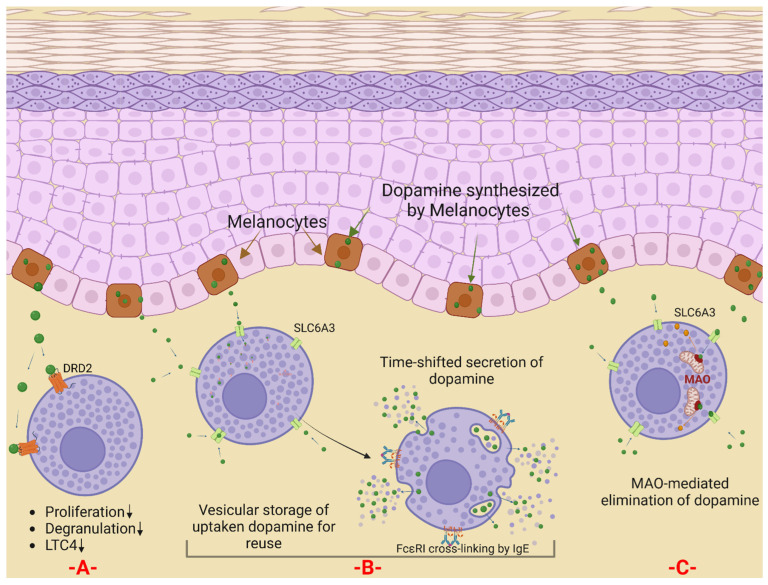
Hypothetical representation of the connections between dopamine and skin mast cells. Some skin cells, including melanocytes, can produce dopamine under physiologic and/or pathological conditions. (**A**) By virtue of their DRD2 receptor, skin MCs can respond to dopamine, which, analogously to other MCs, may dampen their proliferation and function. (**B**) MCs can take up dopamine via SLC6A3. The exogenously provided dopamine can be transported to MC granules and be exocytosed upon appropriate stimulation later. (**C**) Upon dopamine uptake via SLC6A3, dopamine may be transported to mitochondria to be degraded by MAO-A and/or MAO-B. In this latter scenario, MCs would function as dopamine scavengers.

**Figure 5 ijms-23-10871-f005:**
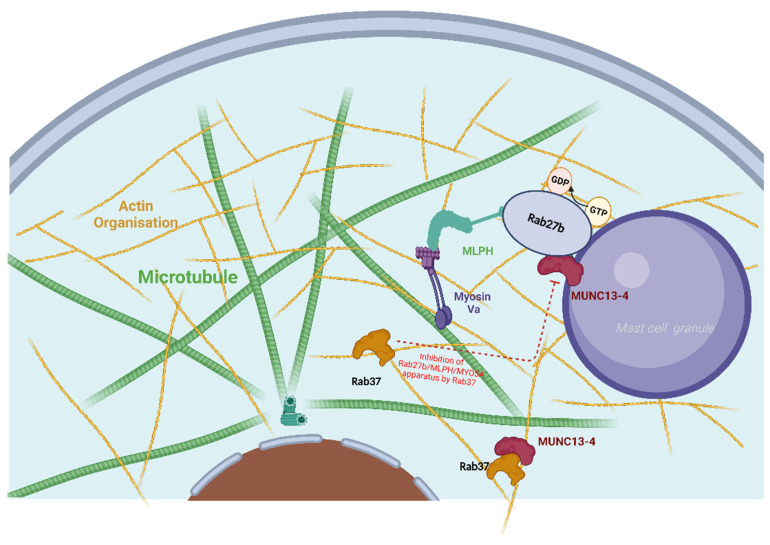
The granule transport machinery of MCs with involvement of melanophilin and Rab proteins. The intracellular transport of cytoplasmic granules to the MC surface for exocytosis requires a complex motor apparatus connected to the granule membrane, in which a motor protein myosin Va and microtubules work together. In this machinery, melanophilin (MLPH) directly connects myosin Va with Rab proteins located on the granule membrane. Subsequent docking to the plasma membrane is carried out by the protein unc-13 homolog D, which in turn can be blocked by RAB37. See main text for further details. Abbreviations: MLPH (melanophilin), MUNC13-4 (Protein unc-13 homolog D), RAB27A (Ras-related protein Rab-27A), RAB27B (Ras-related protein Rab-27B), and RAB37 (Ras-related protein Rab37).

**Figure 6 ijms-23-10871-f006:**
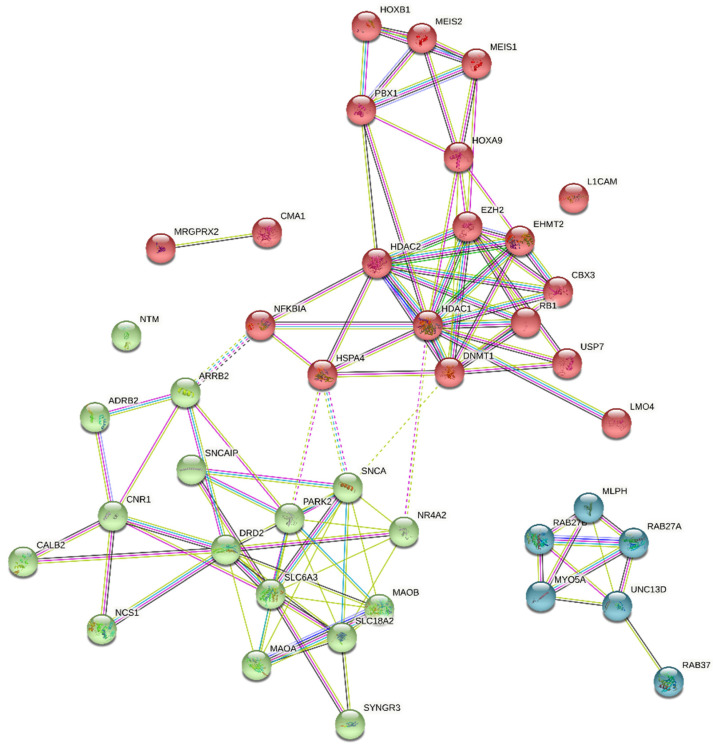
Protein–protein interaction network from the String Database. To predict the functional interactions of selected neuronal genes with other genes transcribed in MCs, a network analysis was performed using the STRING database (https://string-db.org) [242]. Up to the fourth step, additional nodes were added to the basal model until interconnectivity was achieved. The final network is displayed as a specified number (kmeans = 3) of clusters according to its topological properties, with each cluster highlighted in a different color.

**Figure 7 ijms-23-10871-f007:**
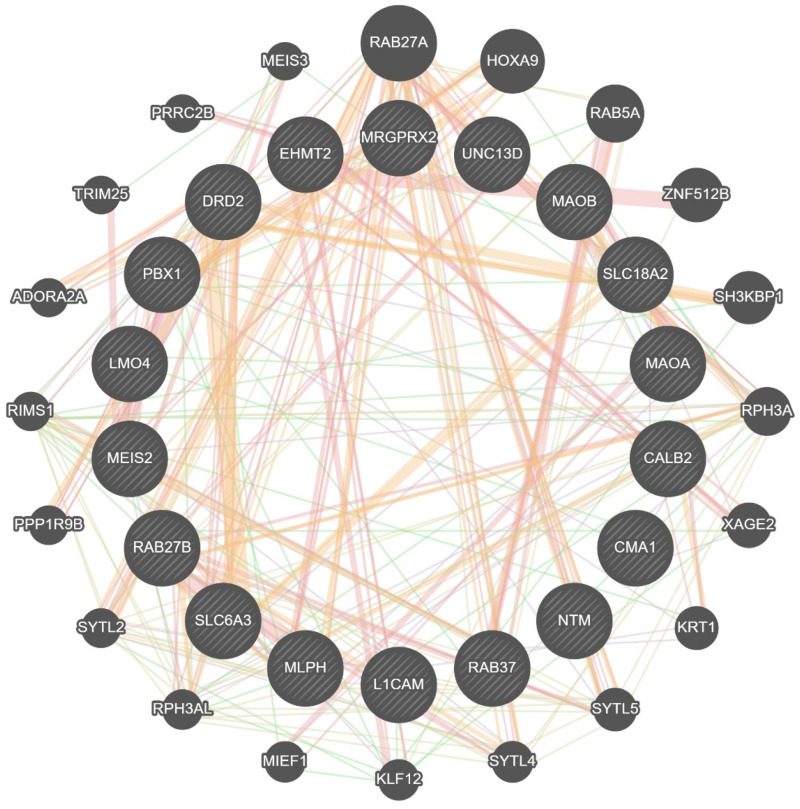
Protein–protein interaction network based on the GeneMANIA web server. To find putative interactions of selected neuronal genes with other partners, network analysis was performed using the GeneMANIA web server [246]. Each gene is a node and evidence for interactions between protein pairs are edges. The thickness of the edges indicates the strength of the interaction, the color of the edges indicates the type of interaction, and the size of the nodes distinguishes the hub gene (inner circles) from the predicted genes (outer circles). Several of the predicted genes are transcribed in MCs (e.g., ADORA2A, SYTL4, KRT1, and MIEF1, and some at substantial levels, e.g., SYTL2).

**Table 1 ijms-23-10871-t001:** Expression of selected gene sets and their comparison between mast cells and myelocytes or basophils. The expression data were extracted from Motakis et al. (2014) [49] and presented as RLE-normalized counts. Here, gene expression of skin-derived MCs was studied in parallel with ~400 different cell types from different tissues, using CAGE-seq analysis. This study provided the most comprehensive overview of the human MC transcriptome ever reported. Fold changes of differentially expressed genes were taken from Babina et al. (2014) [76]. Hereby, the MC transcriptome was compared with different blood cell types from the FANTOM5 project combined (i.e., not only MCs versus a single cell type, but MCs versus an overall myelocytic signature). CMA1 and MRGPRX2 are not discussed herein but were added for comparison as MC-selective entities. RLE Counts: Relative Log Expression (RLE)-based normalized counts by the means of samples.

Gene Names	Mast Cells (Ex Vivo)	Mast Cells (Expanded)	FANTOM5 without MCs	FANTOM5	MC vs. All Myelocytes	MC vs. Basophils
	Mean expression (RLE Counts)	Mean expression (RLE Counts)	Mean expression (RLE Counts)	Mean expression (RLE Counts)	Fold Change (log2)	Fold Change (log2)
DRD2	56.46	54.56	2.62	3.17	10.50	10.11
MAOA	1.78	1401.76	42.60	52.07	- ^1^	- ^1^
MAOB	350.48	403.13	13.76	16.92	13.13	12.40
SLC18A2	786.57	853.81	1.62	8.45	12.49	10.26
SLC6A3	2.38	0.07	0.17	0.18	- ^1^	- ^1^
MLPH	359.39	357.62	29.65	32.29	13.17	11.30
RAB27B	808.76	1484.31	17.67	28.00	7.73	8.64
RAB37	167.54	302.52	12.54	14.32	- ^1^	- ^1^
CALB2	340.87	525.93	22.44	26.35	13.09	12.65
L1CAM	256.85	27.24	38.65	39.36	10.10	9.26
NTM	436.68	96.94	31.09	32.68	13.46	11.40
LMO4	513.30	279.22	126.02	128.08	3.96	3.90
PBX1	53.60	174.26	48.84	49.52	6.63	7.27
MEIS2	129.54	146.93	36.47	37.30	7.78	8.95
EHMT2	41.42	34.74	26.60	26.62	1.63	2.38
CMA1	1668.56	654.30	0.16	9.45	15.39	15.39
MRGPRX2	982.88	85.51	0.04	3.82	14.62	14.38

^1^ Fold changes in expression between cell types are not reported if their difference is not statistically significant.

**Table 2 ijms-23-10871-t002:** Protein abundance of selected proteins in the proteomics study by Plum et al., 2020 [76]. In this study, the proteome of primary human (and mouse) MCs was analyzed by quantitative mass spectrometry. The MC proteome from two compartments (skin and fat tissue) was compared with that of peripheral blood mononuclear cells (PBMCs). Protein abundance was represented as log2 LFQ (Untargeted label-free quantitation) score. Higher LFQ values indicate higher expression of the mentioned protein in the indicated samples. CMA1 and MRGPRX2 are not discussed herein but were added for comparison as MC-selective entities.

Gene Names	Fat Mast Cells (Ex Vivo)	Skin Mast Cells (Ex Vivo)	PBMCs
DRD2	- ^1^	- ^1^	- ^1^
MAOA	- ^1^	- ^1^	- ^1^
MAOB	28.34	28.72	21.89
SLC18A2	21.04	22.81	14.11
SLC6A3	- ^1^	- ^1^	- ^1^
MLPH	22.61	22.01	- ^1^
RAB27B	26.41	26.80	24.53
RAB37	24.39	24.48	20.69
CALB2	25.32	25.98	16.91
L1CAM	23.92	22.87	15.13
NTM	24.28	23.76	18.05
LMO4	- ^1^	- ^1^	- ^1^
PBX1	- ^1^	- ^1^	- ^1^
MEIS2	- ^1^	- ^1^	- ^1^
EHMT2	- ^1^	- ^1^	- ^1^
CMA1	31.22	30.99	21.26
MRGPRX2	20.86	19.63	- ^1^

^1^ “-” indicates that the protein of the corresponding gene was not identified in the mentioned cell types.

**Table 3 ijms-23-10871-t003:** Protein abundance of selected proteins in the proteomics study from our group (manuscript in preparation). The Global proteome in skin mast was analyzed by LC-MS/MS. Protein abundance was represented as log2 LFQ (Untargeted label-free quantitation) score. Higher LFQ values indicate higher expression of the mentioned protein in the indicated samples. CMA1 and MRGPRX2 are not discussed herein but were added for comparison as MC-selective entities. Note that SCF stimulation was performed for a short time only (to detect phosphoproteins); therefore, no changes against expanded baseline cells were expected in accordance with the findings. Thus, this table does not intend to demonstrate regulation by SCF over the observation period. Conversely, it aims to show the good correspondence of the two independent samples to underline robustness of the measured LFQ values.

Gene Names	Mast Cells (Expanded)	Mast Cells (Expanded + SCF Stimulated 30 min)
DRD2	- ^1^	- ^1^
MAOA	34.20	34.00
MAOB	33.32	33.54
SLC18A2	30.70	30.43
SLC6A3	24.65	24.74
MLPH	31.05	31.06
RAB27B	34.80	35.00
RAB37	29.20	29.13
CALB2	32.55	32.42
L1CAM	28.72	28.73
NTM	25.59	26.04
LMO4	- ^1^	- ^1^
PBX1	24.49	24.36
MEIS2	24.65	24.74
EHMT2	27.79	27.64
CMA1	33.59	33.60
MRGPRX2	21.59	22.01

^1^ “-” indicates that the protein of the corresponding gene was not identified in the mentioned cell types.

**Table 4 ijms-23-10871-t004:** Protein abundance of selected proteins in the proteomics study by Dyring-Andersen et al., 2022 [79]. In this study, a spatially resolved quantitative proteome atlas of the different layers and cell types of healthy human skin was created, yielding almost 11,000 proteins, using tissue sections, flow cytometry, and mass spectrophotometry. Protein abundance was represented as log_2_ LFQ (untargeted label-free quantitation) score. Higher LFQ values indicate higher expression of the mentioned protein in the indicated samples. MCs are highlighted by bold print.

Gene Names	Dendritic Cells (CD1+)	Dermal Dendritic Cell	Fibroblast	Keratinocyte	Macrophage	Mast Cell	Melanocyte
DRD2	- ^1^	- ^1^	- ^1^	- ^1^	- ^1^	**- ^1^**	- ^1^
MAOA		28.76	29.02	32.22	29.13	**30.49**	29.14
MAOB	26.80	26.57	26.28	27.35	29.48	**35.71**	28.01
SLC18A2	- ^1^	- ^1^	21.03	21.23	0.00	**30.55**	23.26
SLC6A3	- ^1^	- ^1^	- ^1^	- ^1^	- ^1^	**- ^1^**	- ^1^
MLPH	25.75	26.50	28.94	25.49	26.74	**31.90**	31.83
RAB27B	23.66	26.58	25.03	28.94	25.18	**35.20**	29.12
RAB37	25.20	26.77	25.56	26.04	25.67	**32.37**	27.69
CALB2	26.47	27.71	28.64	29.19	26.69	**34.81**	26.83
L1CAM	- ^1^	22.69	21.48	27.31	21.96	**26.08**	27.17
NTM	22.75	22.34	27.12	24.68	23.67	**26.19**	23.67
LMO4	- ^1^	- ^1^	- ^1^	- ^1^	- ^1^	**- ^1^**	- ^1^
PBX1	- ^1^	- ^1^	- ^1^	- ^1^	- ^1^	**- ^1^**	- ^1^
MEIS2	22.60	-	28.55	22.33	- ^1^	**25.20**	23.84
EHMT2	28.09	26.93	28.48	28.03	27.30	**27.62**	26.35
CMA1	29.70	29.81	29.07	29.97	33.57	**38.39**	29.16
MRGPRX2	- ^1^	- ^1^	- ^1^	- ^1^	21.97	**30.66**	-

^1^ “-” indicates that the protein of the corresponding gene was not identified in the mentioned cell types.

**Table 5 ijms-23-10871-t005:** Gene expression values were extracted from the ImmGen database [224]. In the associated study [80], constitutive MCs from five tissues were analyzed using microarray-based gene expression analysis. Gene expression in basophils and eosinophils from two compartments each were likewise quantified. Expression is presented as RMA-normalized signal intensity for probe of corresponding gene (not log-transformed). (RMA = Robust Multichip Average, an approach to normalize each array against all others).

Gene Name	Tracheal Mast Cells	Tongue Mast Cell	Dermal Mast Cell	Peritoneal Mast Cell	Esophageal Mast Cells	Adipose Tissue Eosinophils	Peripheral Blood Eosinophils	Splenic Basophils	Blood Basophil
Drd2	80	90	80	78	84	115	107	102	89
Maoa	69	79	108	65	90	47	51	45	44
Maob	4195	4654	5366	4602	3805	47	50	48	55
Slc18a2	4461	4387	3749	4547	3676	121	109	1981	1960
Slc6a3	138	145	127	134	150	183	144	145	131
MLPH	1951	1813	1424	1874	1197	220	185	206	198
Rab27b	3591	3586	3209	3503	3447	65	61	73	85
Rab37	1153	1223	1113	1544	788	742	692	1542	1489
Calb2	42	44	40	44	44	57	56	46	45
L1cam	275	625	308	599	590	451	679	2338	3104
Ntm	67	77	71	62	72	101	77	74	69
Lmo4	1018	890	1026	1100	784	549	1521	2095	2021
Pbx1	566	471	438	430	410	129	149	326	356
Meis2	3953	2786	2910	4481	2819	85	64	52	53
Ehmt2	324	335	388	388	306	267	283	440	384
Cma1	4485	4452	3201	4694	3814	57	57	63	55
Mrgprb2	3619	3497	2881	4380	3465	18	24	16	18

RMA normalized values, microarray data from Immgen Mouse Database [80].

## Data Availability

No datasets were generated during this study. This study analyzed previously published datasets [49,72,76,78,79,82].

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
