# Peer review of "How “Neuronal” Are Human Skin Mast Cells?"

_ijms, 2022, doi:10.3390/ijms231810871_

Round 1
Reviewer 1 Report
Please see attached file.

Reviewer 2 Report
In this manuscript, Babina M et al review the information related to specific gene transcription in mast cells, specially the obtained with high-throughput technology, in order to identify groups of genes of neuronal markers specifically hyper-expressed in MCs, in comparison with other myeloid cells. Analyzing data reported by themselves and other groups, they identified genes belonging to several subcategories, such as traffic/lysosomes, the dopaminergic system, calcium-regulated proteins, adhesion molecules and transcription factors and modulators of transcriptional activity. Then, they discuss discuss the similarities between CNS and mast cell gene expression, since MCs share functions with the nervous system (i.e. they produce some neurotransmitters MC degranulation machinery shares features with that found in the synapsis).
This is an interesting review that contributes to the analysis of the gene signature of some types of MCs, placing them as a particular cell lineage with very unique characteristics.
Some aspects must be attended.
1. Since MCs constitute a very heterogeneous cell type, able to alter their phenotype depending the conditions and the tissues where they reside, results of gene expression in distinct populations may importantly vary. In this manuscript, authors discuss mainly studies performed with human skin mast cells and, in distinct points, they analyze the function of selected proteins and genes in the context of cutaneous environment (see, for example, lines 397-399 and 931-932). Also, they make reference to the ancient MCs found in the sea squirt Ciona intestinalis, that resembles cutaneous human MCs since they store histamine and heparin-serine protease complexes in their granules and they respond to compound 48/80 (Wong, et al., BBRC 451:314-318, 2014). Authors should consider modify the title to “How neuronal are human skin mast cells?”
2. Authors must explain with more detail the experimental design of the whole study and methods used to perform the analysis showed in Tables 1, 2 and 3. Besides, Table 3 shows results of stimulated during 30 minutes. ¿Do the authors suggest that neuronal genes expressed in MCs are inducible under certain circumstances? Please explain.
3. In lines 52-64 authors mention that, since MCs appeared before B cells and IgE, the lineage is believed to have important roles of non-IgE drive processes. As they also mention, a number of studies strongly suggest that ancient functions of MCs are related to innate immunity responses and defense (Reviewed in Saccheri, P., et al., IJAE, 124:271-287, 2019). How the new evidence on the expression of nervous system genes in MCs changes this view? How the authors insert their findings on the knowledge of the role of MCs in innate immunity?
4. In the Summary, future perspectives and concluding remarks, authors should state their main conclusions and justify the question they use as the title of their study. For example, does they mean that a cell that communicates with neurons through the secretion of distinct mediators and express some groups of genes also found in neurons should be recognized as “neuronal”? What does “neuronal” means for the authors? Secretory cells, such as pancreatic beta cells or chromaffin cells of the adrenal medulla also express a number of genes of enzymatic and secretory pathways related to those found in neurons, should they also be considered “neuronal”?
Minor comment:
5. Line 49 says 500 years and it should say 500 million years.
